# Farming, Pesticides, and Brain Cancer: A 20-Year Updated Systematic Literature Review and Meta-Analysis

**DOI:** 10.3390/cancers13174477

**Published:** 2021-09-05

**Authors:** Nicole M. Gatto, Pamela Ogata, Brittany Lytle

**Affiliations:** School of Community and Global Health, Claremont Graduate University, Claremont, CA 91711, USA; pamela.ogata@cgu.edu (P.O.); brittany.lytle@cgu.edu (B.L.)

**Keywords:** farming, agriculture, pesticides, brain cancer, glioma, meta-analysis, systematic literature review, evidence synthesis

## Abstract

**Simple Summary:**

A synthesis of 40 years of epidemiologic studies of farming and brain cancer that updates two previous meta-analyses finds that farming with its potential for exposure to chemical pesticides is associated with an increased risk of brain cancer.

**Abstract:**

Twenty additional years of epidemiologic literature have become available since the publication of two meta-analyses on farming and brain cancer in 1998. The current systematic literature review and meta-analysis extends previous research and harmonizes findings. A random effects model was used to calculate meta-effect estimates from 52 studies (51 articles or reports), including 11 additional studies since 1998. Forty of the 52 studies reported positive associations between farming and brain cancer with effect estimates ranging from 1.03 to 6.53. The overall meta-risk estimate was 1.13 (95% CI = 1.06, 1.21), suggesting that farming is associated with a 13% increase in risk of brain cancer morbidity or mortality. Farming among white populations was associated with a higher risk of brain cancer than among non-white populations. Livestock farming (meta-RR = 1.34; 95% CI = 1.18, 1.53) was associated with a greater risk compared with crop farming (meta-RR = 1.13; 95% CI = 0.97, 1.30). Farmers with documented exposure to pesticides had greater than a 20% elevated risk of brain cancer. Despite heterogeneity among studies, we conclude that the synthesis of evidence from 40 years of epidemiologic literature supports an association between brain cancer and farming with its potential for exposure to chemical pesticides.

## 1. Introduction

While overall cancer death rates in the United States (US) have declined between 1999 and 2018, deaths from primary cancers of the brain and central nervous system (CNS) have increased during 2014 and 2018 [1]. In 2019, brain cancer was the 10th leading cause of cancer death in both men and women overall [2]; among those aged 20 to 39 years, it was the leading cause of cancer death among men, and the fourth leading cause of cancer death among women [2]. Despite advances in medicine, treating brain cancer is challenging because of the out-of-reach locations of tumors, the natural defenses from the blood–brain barrier, outward extensions of primary tumors into other parts of the brain, and the existence of multiple mutations within tumors [3]. For these and other reasons, the five-year survival rate for brain cancer (33%) is lower than many other cancers [4]. More research on the etiology and risk factors is needed to inform primary prevention strategies.

The epidemiology of primary brain cancers, which include meningiomas and gliomas, is largely inconclusive [5,6]. In the US, glioma incidence is higher in men and is more common among white compared with black persons. Meningioma incidence is higher in women, and tends to be diagnosed at older ages [6]. While numerous risk factors for both gliomas and meningiomas have been investigated, evidence is best established for ionizing radiation, family history of brain cancer, and certain hereditary syndromes [5]. Brain cancer incidence further varies by geography [4], suggesting that environmental factors could contribute to risk. Many workplace environments involve exposures to carcinogens and neurotoxic substances that cause brain tumors in experimental animals [7], raising the possibility that certain occupations and industries may be at greater risk.

The agriculture industry, and specifically farming occupations, utilize chemical pesticides and fertilizers, many of which have been assessed for their potential as human carcinogens [8,9]. Concerns about possible brain cancer risk from exposure to such chemicals have led to numerous epidemiologic studies of farmers [10,11,12,13,14,15,16,17,18,19] with two meta-analyses conducted in the 1990s [20,21]. The first meta-analysis of 33 studies reported a meta-relative risk estimate of 1.30, 95% confidence interval (CI) = 1.09, 1.56 [21]; the second meta-analysis, which included 28 studies, calculated a meta-relative risk estimate of 1.06, 95% CI = 1.02, 1.11 [20]. The authors of the former concluded that the meta-analysis supported an association between farming and brain cancer; the authors of the latter concluded that there was no clearly elevated risk of brain cancer among farmers.

As the world’s population is projected to double by 2050, farmers are expected to continue to use pesticides as important tools among available technologies [22] to control pests and achieve needed increases in production [23]. Pesticide products, formulations, and application practices have changed with time [24], and additional epidemiologic studies of farmers have been conducted since the two previous meta-analyses were published. Thus, the current systematic literature review and meta-analysis aims to extend previous research by twenty years to include epidemiologic studies published since 1998, and to harmonize findings from the previous meta-analyses on studies of farming and brain cancer.

## 2. Materials and Methods

### 2.1. Literature Search

Using PubMed and Agricola databases, a comprehensive literature search was conducted to identify epidemiologic studies evaluating the relationship between farming and brain cancer that were published in English between 1 January 1997 and 1 August 2019 in order to locate studies that had been published after the two previous meta-analyses [20,21]. The following keyword search terms were used: “farming and brain cancer”, “farm and cancer”, “farming and cancer”, and “farmers and cancer.” Abstracts of studies were obtained and reviewed to ascertain whether inclusion criteria were met; if this could not be determined by the abstract, the full text was used. We also attempted to locate all studies that were included in the two previous meta-analyses [20,21].

### 2.2. Inclusion Criteria

Studies were retained if they met the following criteria:Epidemiologic studies of adult populations that included farmers (i.e., farmers, farm managers, agricultural workers, and/or wives of farmers).Morbidity or mortality from brain cancer or cancers of the central nervous system was a reported outcome.Measures of effect were estimated in the study, or data were available that allowed for the calculation of a relative risk estimate (i.e., Standardized Incidence Ratio (SIR), Standardized Mortality Ratio (SMR), Proportionate Mortality Ratio (PMR), Relative Risk (RR), Odds Ratio (OR), Proportionate Cancer Mortality Ratio (PCMR), Mortality Odds Ratio (MOR)) and a measure of variability (i.e., 95% confidence interval (CI)).

When multiple papers were available on the same study population (i.e., a cohort had been updated over time), the most recently published article or most comprehensive analysis on brain cancer in farmers was selected.

### 2.3. Extracting Data from Included Studies

Two authors (BL and PO) independently reviewed each study to abstract data using a standard format. Information was collected and summarized on each study as follows: publication year (or date of report if not published); source identified (Khuder et al., 1998 [21] or Acquavella et al., 1998 [20] meta-analysis; PubMed or Agricola database search); geographic region (Europe including the United Kingdom, Scandinavia, and Iceland; US; Canada; China; Brazil; New Zealand; International) and location(s); epidemiologic design; population included; type of farmer; number of exposed brain cancer cases or deaths; measure of association; and inclusion of subcohort(s) with potentially higher exposure (either based on duration of employment or job tasks that would result in greater pesticide exposure). For type of farmer, studies were classified based on how effect estimates were reported: crop only; livestock only; livestock and crop (both included; reported separately); mixed (both included but reported together in a single estimate); or unspecified (could not be determined from information provided). Measures of effect and variability were extracted for brain cancer, and, if available, brain cancer among a more highly exposed subcohort, and all cancer. If effect measures were not reported, available data in the study were used to calculate them. Articles were frequently discussed individually to reconcile differences if abstracted data varied between authors.

Several procedures were applied to the data abstraction process as follows.

When a study presented risk estimates (i.e., SMRs) for more than one reference group, such as with a national population and a regional population, we used the latter.When risk estimates were presented for both industry (i.e., the type of activity at a person’s work) and occupation (i.e., the kind of work a person does to earn a living), we used the latter and, when available, selected risk estimates associated with “farming” as an occupation. If risk estimates for multiple farming occupations (i.e., farmers, farm managers, farm laborers) were presented, we pooled estimates and, when possible, abstracted those which specifically attributed to crop (i.e., corn, peanuts, grains, tobacco) or livestock (i.e., poultry, cattle, hogs) farming. When a study made distinctions between “farmers” and “other agricultural occupations” or “forestry workers and fishermen” in their reporting of risk estimates, we extracted those for “farmers”.When risk estimates were reported for outcomes that included “diseases of the nervous system”, “all nervous system cancers”, and “brain cancer”, we extracted those specific to “brain cancer”. When studies reported separate risk estimates for more than one histological type of brain cancer, or for malignant and benign neoplasms, we pooled estimates across the brain cancer subtypes.When a study reported more than one measure of risk (i.e., SMR, OR), we prioritized our abstraction as follows: RR, OR, SMR/SIR, PMR/PCMR. If a study reported crude and adjusted risk estimates, we used adjusted measures. If 95% confidence intervals were not reported and approximate CIs could not be calculated with data provided by the study, we used the 90% CIs.When a study reported effect measures for two (or more) separate population cohorts and reported risk estimates for these subcohorts individually, we used these risk estimates as separate observations, i.e., as if they were two (or more) studies. If the study did not provide data to be able to calculate effect measures on the entire cohort, we used the data for the portion of the cohort that was available in our calculations.

We handled other alternatives that arose through these rules using a series of sensitivity analyses (see below), which allowed us to examine the influence on the meta-effect measure by including individual effect measures calculated with different reference populations or adjusted for covariates.

### 2.4. Evaluation of Quality

Two authors (N.M.G. and P.O.) subjectively evaluated the quality of each study based on the completeness of the methods described and results presented. We considered how well farming and exposures were characterized, whether studies provided specific industry or occupation codes, and how historical employment was addressed. In addition, we evaluated how brain cancer was defined (e.g., CNS cancer, malignant brain cancer), how cases were identified (via self-report, registry, death certificate), and whether there was histological confirmation of the cancer. Considered were whether and what potential confounding factors were accounted for and what steps were taken to avoid biases. We assessed how adequately results were presented, whether data were missing, and whether it was possible to obtain data from the report for calculations, if needed.

A 26-point quality assessment tool (Appendix A, Table A1) was developed to evaluate studies, modeling after the Newcastle–Ottowa quality assessment scale for case-control studies and using STROBE guidelines [25,26]. The inter-rater reliability of the tool was high (r = 0.69, *p* < 0.0001), with scores ranging from 10 to 25 for reviewer one, and 9 to 23 for the second reviewer. Continuous scores were classified into tiers (poor, fair, good) by dividing the range to create tertile cutpoints for the categories. Both authors re-assessed articles for which rater tier scores differed from each other. Agreement between authors in quality tier scores was reached after a second review and discussion.

### 2.5. Meta-Analysis Model

We used a random effects model to calculate pooled, meta-effect estimates (as approximations of meta-RRs), which assumes that the study-specific effect sizes come from a random distribution of effect sizes according to a specific mean and variance [27]. For the calculations, risk estimates from individual studies were weighted by the inverse of the variance, which accounts for the size of study populations. Meta-effect estimates were calculated for groups and subgroups of three or more studies.

### 2.6. Analysis of Heterogeneity and Influence and Sensitivity Analyses

We quantitatively assessed variability in pooled estimates across studies using a *p*-value < 0.10 from tests of homogeneity to suggest an investigation into potential sources of heterogeneity. As data permitted, we explored potential sources of variability by calculating meta-effect estimates for subgroups within the studies based on: geographic region where the study was based (the US, Europe/United Kingdom, Canada); type of study design (cohort, case-control, PMR); year of study publication (prior to 1990, between 1990 and 1999, 2000 or later); sex (male, female); race (white, non-white); type of farmer (livestock only; crop only); farming duration (less than 10 years, 10 years or more); type of brain cancer (glioma); source of exposure classification (personal interview or self-report; registry or administrative list including those from agricultural organizations, agricultural census; death certificate); and quality assessment tier (good, fair, poor).

In sensitivity analyses, we calculated meta-effect estimates for studies that specifically reported data for a more highly exposed subcohort of workers and for farmers who had documented exposure to pesticides.

We examined the relative influence of specific studies or combinations of studies by calculating pooled estimates with the removal of a study(ies) and comparing them to the overall estimate.

We used the Preferred Reporting Items for Systematic reviews and Meta-Analyses (PRISMA) flow diagram to report identified records, excluded articles, and included studies [28]. All analyses were performed using Episheet Statistical Software [29], a spreadsheet-based analytical package designed for the analysis of epidemiologic data. Institutional Review Board review was not obtained because the study was not human subjects research.

## 3. Results

Of the 61 total studies covered by the two previous meta-analyses, 16 were included in both reviews. Two articles [30,31] from Acquavella could not be located despite efforts to contact the original authors or through extensive efforts of a librarian. Once reviewed for eligibility, one article [32] did not have usable data on brain cancer. An additional study in Acquavella [33] was updated by a study [34] in Khuder. Similarly, two articles [35,36] in the Khuder meta-analysis were updated by other studies [37,38]. The literature search identified 143 studies in PubMed and 144 in Agricola. After screening, 245 of these were determined to be duplicates or not epidemiologic studies and thus were not pursued. Further review led to the exclusion of 31 studies for not meeting inclusion criteria or for reporting data that had been covered by more recent studies. One published article [39] reported risk estimates from separate analyses among brain cancer cases in a Danish cohort of farmers and brain cancer deaths in an Italian cohort of farmers; we counted this as two studies. In total, 52 studies from 51 articles or reports [10,11,12,13,14,15,16,17,18,19,34,37,38,39,40,41,42,43,44,45,46,47,48,49,50,51,52,53,54,55,56,57,58,59,60,61,62,63,64,65,66,67,68,69,70,71,72,73,74,75,76] were included in the current meta-analysis, including 11 additional studies that were published after the two previous meta-analyses (Figure 1, Table 1).

All but three of the articles in the current meta-analysis included study populations from North America or the European region, including the United Kingdom, Scandinavia, and Iceland. Twenty cohort studies, 20 case-control studies, and 12 PMR studies contributed data from predominantly white males. Farming occupations were assessed from registries or other administrative lists, death certificates, personal interviews, and self-reports. From the descriptions provided, the type of farmer could be determined for 31 (60%) studies, with 16 of these providing data on crop farming and 13 for livestock. Four studies reported data on duration of farming, 19 included data for one or more highly exposed subcohorts, and eight provided risk estimates specifically for glioma brain tumors. The majority (77%) of studies were judged to be of fair or poor quality (Table 2).

Forty of the 52 (77%) studies reported positive associations between farming and brain cancer, with effect estimates ranging in magnitude from 1.03 to 6.53. Of the 11 studies that reported negative associations, only three had estimates that were different than the null (Table 2). These three studies were a mixture of cohort, case-control, and PMR studies conducted in the US, Europe, and Canada among farmers of unspecified types and published in 1992 [39,49] and 2006 [63].

The overall meta-risk estimate for all 52 studies combined was 1.13 (95% CI = 1.06, 1.21), suggesting that farming is associated with a 13% increase in risk of brain cancer morbidity or mortality. However, the studies combined in this pooled analysis were significantly heterogeneous (*p*-homogeneity < 0.0001). The elevated risk was apparent and did not vary by the publication date of studies, either prior to 1990, between 1990 and 1999, or in the year 2000 or later, as indicated by overlapping confidence intervals of these subgroup risk estimates. The meta-effect estimate calculated for US studies (meta-RR = 1.13; 95% CI = 1.03, 1.23) was comparable to the overall estimate. The risk of brain cancer from farming in Europe and the United Kingdom (meta-RR = 1.08; 95% CI = 0.98, 1.17) or Canada (meta-RR = 0.92; 95% CI = 0.66, 1.26) was not different than the null. Among the epidemiologic study designs, the pooled estimate from cohort studies did not indicate an elevated risk, whereas, in case-control studies, the meta-effect estimate was 1.39 (95% CI = 1.10, 1.75). Results did not indicate a difference in the effects of farming on brain cancer between men and women or by the source with which farming exposure was classified. Farming among white populations was associated with a higher risk of brain cancer than among non-white populations (Table 2).

Livestock farming (meta-RR = 1.34; 95% CI = 1.18, 1.53) was associated with a greater risk of brain cancer than crop farming (meta-RR = 1.13; 95% CI = 0.97, 1.30). The risk of a glioma tumor type was not statistically significantly elevated (meta-RR = 1.11; 95% CI = 0.95, 1.31). The four studies [13,15,50,76] that allowed for an examination of farming duration were published in the 1980s, 1990s, and 2000s among populations in Europe and the US and were all case-control studies. Meta-risk estimates for both farming with a duration less than 10 years and 10 years or more were elevated, with the latter associated with a 72% elevated risk of brain cancer. However, one of the studies in this analysis [50] included only 10 cases. Removing this study from the pooled estimate resulted in a meta-SMR = 1.87 (95% CI = 1.19, 2.96). Studies that were evaluated as being of good quality had a pooled risk estimate of 1.40 (95% CI = 1.09, 1.81). Those of fair or poor quality did not support an association between farming and brain cancer risk.

Nineteen studies included data for a more highly exposed subcohort; the meta-effect estimate (1.15; 95% CI = 1.0, 1.32) for these studies was similar in magnitude to the overall estimate and statistically significantly elevated. Ten studies provided documentation to substantiate that farmers had been exposed to pesticides. The pooled estimate for these studies was 1.22 (95% CI = 1.10, 1.35).

Despite the various subgroup analyses, significant heterogeneity remained in the majority (88%) of the pooled estimates.

## 4. Discussion

This comprehensive review and meta-analysis encompassing 42 years of the epidemiologic literature and updating two previous meta-analyses by 20 years supports an association between farming and brain cancer incidence and mortality. The magnitude of the overall meta-risk estimate for 52 studies is between that of the Khuder et al. 1998 [21] meta-analysis (1.30) and the Acquavella et al. 1998 [20] meta-analysis (1.06), all three of which reflect statistically significant elevations in brain cancer risk from farming. Our analyses suggest that the elevated risk has been consistent over time and the addition of newer studies (i.e., those published since 2000) does not change this conclusion.

In the Khuder et al. meta-analysis [21], the meta-effect estimates calculated for studies that were based on death certificate data or US-based studies were comparable to the overall estimate for the 33 total studies they included. Similarly, the meta-effect estimates that we calculated for studies that used death certificates as the source of exposure classification or those that were based in the US were not different than the overall meta-risk estimate for the 52 studies that we included. Our results for female farmers generally agree with those by Khuder et al., [21] who found smaller non-statistically significantly elevated risks compared with male farmers.

In the Acquavella et al. meta-analysis [20], the results by epidemiologic study design follow a different pattern than our study. In their study, pooled associations were strongest for PMR studies, followed by “follow-up” studies and then case-control studies. We, on the other hand, found meta-effect estimates to be of greatest magnitude for case-control studies followed by PMR studies followed by cohort studies. One explanation for this may be the publication of additional, higher-quality case-control studies after 1998 that reported positive associations. The estimated meta-effect for cohort studies in our meta-analysis was of similar magnitude to that in Acquavella, but unlike Acquavella it had evidence of heterogeneity. In line with Acquavella et al. [20], we found that the pooled estimate for PMR studies did not deviate from homogeneity.

Meta-analyses are a quantitative method to derive a more statistically precise risk estimate and a better understanding of the consistency (or inconsistency) of findings in the literature [27]. Additionally, meta-analyses frequently make it possible to investigate whether there are patterns in the exposure–disease relationship based on subgroups within a population for which individual studies may not provide sufficient numbers. We calculated meta-RRs for a number of different subgroups so that we could more closely examine if the effect of farming on brain cancer varied by characteristics of the farmer among others. Our results suggest that regardless of whether analyses were stratified by the sex of the farmer, the source of exposure classification, or were specific to non-white populations or crop farmers, the observed association between farming and brain cancer was consistent. Taken together, the consistency across different subpopulations contributes evidence to substantiate that the association identified between farming and brain cancer may be causal. Furthermore, while the number of studies with data on duration of farming were few, our results suggest a dose–response relationship such that more years farming was associated with a greater risk of brain cancer. Yet, we did not observe a greater risk among more highly exposed subcohorts. While cohort studies are a stronger observational study design, our subgroup analyses of cohort studies did not support an elevated risk as was found in the case-control and PMR studies. Additionally convincing were analyses restricted to good quality studies that showed statistically significant elevations in brain cancer risk. The subgroup analyses were also an approach to reduce the heterogeneity that was apparent in our overall meta-effect estimate. Nevertheless, because heterogeneity persisted in many of our pooled estimates, the results of this meta-analysis should be interpreted with caution.

Farmers may be occupationally exposed to pesticides by applying them to control agricultural pests, including weeds, and to eliminate rodents. Thousands of pesticides, including insecticides, herbicides, and fungicides, are widely used worldwide in agriculture [77]. Pesticide formulations contain more than 800 active chemical ingredients [78], some of which are known mechanistically to cause DNA or chromosomal damage [79]. Several specific pesticides have been evaluated by IARC Working Groups for their potential as human carcinogens and conclusions have varied depending on the pesticide or pesticide class, from carcinogenic to probably carcinogenic to possibly carcinogenic to not classifiable [80]. Currently, however, no pesticide has sufficient or limited evidence in humans to be classified as a brain and or central nervous system carcinogen by IARC [80,81].

In the current meta-analysis, we extended beyond previous work and calculated a meta-estimate for farmers exposed to pesticides. Restricting analyses to the ten studies in our meta-analysis that convincingly documented pesticide exposure among farmers suggested that pesticide use in farming occupations was associated with greater than a 20% elevated risk of brain cancer. The observation that farming–brain cancer associations varied by type of farmer could suggest that exposures, including those to pesticides, are different by crop or livestock farming. Many chemical classes of pesticides used in the US, for example, vary depending on the type of crop (cotton, corn, rice, etc.), and insecticides are commonly used externally and systematically in livestock to control parasites [77]. Differences in farming practices, pesticide purposes and uses, and toxicity of chemicals could underlie our findings of region-specific meta-effect estimates. Indeed, historically the US has been more receptive to chemical pesticide and fertilizer use in farming than in Europe [82].

In addition to pesticides, farmers are also exposed to other potentially hazardous chemical and biological agents, such as solvents, fuels, and oils, biologically active dusts, viral and bacterial exposures from farm animals, nitrates from fertilizers, and other chemicals widely generated in agriculture, such as formaldehyde, ammonia, hydrogen sulfide, and phenols, all which can have adverse health effects in their own right [16,83,84,85]. Therefore, analyses of pesticide exposure among farmers may be confounded by co-exposures to these and other agents. Several [14,34,66,76] but not all [17,51] studies included in our meta-analysis controlled for a limited number of other brain cancer risk factors, such as age, or restricted analyses to farmers of male sex or white race, which would reduce the potential that individual reported risk estimates would be confounded by these factors. However, studies generally did not control for other brain cancer risk factors such as medications, anthropometric factors, or socioeconomic status (SES) [86]. Though it could be argued that farmers are a relatively socioeconomically homogenous group compared with other occupations, many studies included both farm laborers and farm managers, who would not be expected to be of similar SES [42,47,54]. Furthermore, relatively little is known about the etiology of brain cancer [86]. Thus, it is not possible to rule out that estimates from individual studies could be confounded by established and yet to be substantiated factors, implying that the meta-effect estimates would be confounded as well. Finally, long lag times between environmental exposures and the development of cancer make it challenging to reach conclusions about occupational exposures, such as pesticides, and cancer risk.

Gliomas, which arise from the supportive tissue in the brain called the glia, account for 80% of intracranial primary malignant tumors [87]. Our subgroup analyses of the eight studies that presented data specifically for gliomas resulted in a meta-effect estimate similar in magnitude to the overall estimate. Insufficient data were available from included studies to examine other brain cancer types. Furthermore, in many studies, brain and CNS cancers were reported together, so our risk estimates would not be reflective of brain cancer exclusively. Additional case-control studies would be needed to further examine farming and pesticide exposure in different primary brain cancer types.

Publication bias, the propensity to publish studies with positive results but not publish studies with negative or null results, is a consideration in meta-analyses [27]. Both Khuder et al. [21] and Acquavella et al. [20] concluded that publication bias could not be ruled out as a possibility in their analyses. Publication bias is a possibility in our research given that 40 of the 52 studies that we identified for inclusion reported positive associations between farming and brain cancer. However, as noted above, the consistency of evidence across populations, geographic areas, and sexes in over four dozen studies makes it likely that there is a true association between farming and brain cancer that is not a consequence of publication bias.

## 5. Conclusions

In summary, despite heterogeneity among studies, we conclude that the synthesis of evidence from over 40 years of epidemiologic literature supports an increased risk of brain cancer from farming with its potential for exposure to chemical pesticides. Increasing organic farming practices is one means to reduce the exposure of farmers to chemical pesticides.

## Figures and Tables

**Figure 1 cancers-13-04477-f001:**
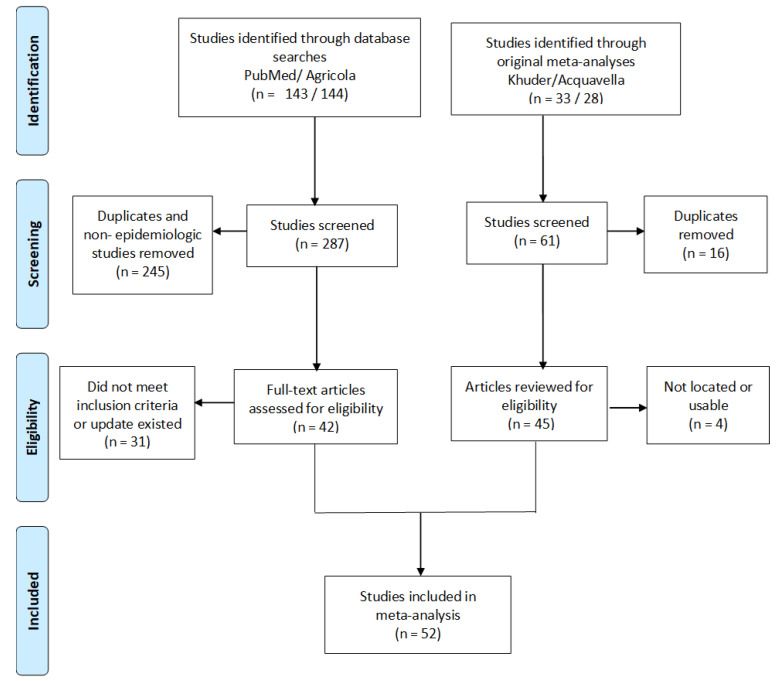
PRISMA [28] flow diagram showing studies considered for inclusion in meta-analyses of farming and brain cancer.

**Table 1 cancers-13-04477-t001:** Summary of studies included in meta-analysis of farming and brain cancer.

First Author	Publication Year	Source of Article	Region (Location)	Study Type	Study Population	Type of Farmer	Subcohort with Higher Exposure	Number of Exposed Brain Cancer Cases or Deaths	Measure of Association	Brain Cancer Risk Estimate (95% CI)	Risk Estimate among More Highly Exposed Sucohort (95% CI)	All-Cancer Risk Estimate (95% CI)
Alberghini [40]	1991	Khuder	Europe (Italy)	Cohort	deaths among 4580 male farmers licensed to buy and use pesticides in the Emilia Romagna region of Italy during 1974–1987; both regional and national populations were used as a reference	crop	no	11	SMR	1.39 (0.69, 2.46)	-	0.68 (0.60,0.78)
Blair [41]	1993	Khuder	US (multiple states)	PMR	deaths in 23 US states during 1984–1988 among farmers (n = 135,560) compared with non-farmers	mixed	no	473	PMR	1.14 (1.04, 1.24)	-	0.87 (0.81,0.93)
Brownson [34]	1990	Khuder	US (Missouri)	Case-control	312 histologically confirmed brain and other CNS cancer cases and 1248 frequency-matched controls (patients with other cancers) in white males from the Missouri Cancer Registry during January 1984–1988	not specified	no	21	OR	1.10 (0.60, 1.70)	-	-
Burmeister [42]	1981	Khuder	US (Iowa)	Cohort	121,101 deaths among white males aged >20 in Iowa during 1971–1978; Iowa population was used as a reference	not specified	no	111	SMR	1.13 (0.66, 1.82)	-	1.12; *p* < 0.01
Cerhan [43]	1998	Pubmed	US (Iowa)	PMR	88,090 deaths among white males 20 years or older in Iowa during 1987–1993; deaths among farmers (n = 5552) were compared to those among non-farmers (n = 82,538)	not specified	age 65+ (assumed longer exposure)	117	PMR	1.10 (0.92, 1.32)	1.09 (0.88, 1.36)	0.92 (0.90, 0.94)
Corrao [44]	1989	Acquavella	Europe (Piedmont, Italy)	Cohort	hospitalizations for a malignant neoplasm among 25,945 male farmers licensed to purchase and use pesticides between 1970 and 1974	crop	cluster of villages with the greatest arable land	25	SIR	1.0 (0.6, 1.4)	0.7 (0.3, 1.3)	0.7 (0.6, 0.8)
Dean [45]	1994	Khuder	Europe (Ireland)	Cohort	deaths caused by brain cancers among farmers in Ireland between 1971 and 1987; the general population was used as a reference	not specified	no	240	SMR	0.91 (0.62, 0.98)	-	-
Decoufle [46]	1977	Acquavella	US (New York)	Case-control	25,416 cancer patients admitted to a cancer treatment center in New York between 1956 and 1965 and non-cancer patients	mixed	no	2	OR	6.53 (0.57, 74.24)	-	-
Delzell [10]	1985	Khuder	US (North Carolina)	PMR	74,041 deaths among black and white male residents of North Carolina during 1976–1978 aged ≥15 years	livestock, crop	level of agricultural activity (non-whites only); age at death ≥65 years	27	SMR	1.21 (0.38, 3.88)	0.91 (0.34, 2.48)	0.90 (0.81, 1.0)
Demers [47]	1991	Khuder	US (Washington)	Case-control (cancer deaths)	white males aged 20 years and older who died in Washington state between 1969 and 1971 from a brain tumor (n = 904) and from a cause other than brain cancer, CNS tumor, or leukemia (n = 904) as identified from death certificates	not specified	farmers and farm managers	63	OR	1.05 (0.75, 1.47)	1.0 (0.7, 1.6)	-
Faustini [48]	1993	Acquavella	Europe (Aprilia, Italy)	Cohort	deaths among 2127 registered farmers aged 30 years or older who were residents of Aprilia between 1971 and 1988; the population of Italy was used as a reference	crop	no	3	SMR	1.38 (0.44, 4.35)	-	0.81 (0.58,1.12)
Fincham [49]	1992	Khuder	Canada (Alberta)	Case-control	1130 male farmers and 3563 men in other occupations between 25 and 74 years of age from the Alberta Cancer Registry during 1983–1988 diagnosed with their first neoplasm	not specified	no	8	OR	0.33 (0.16, 0.68)	-	-
Forastiere [50]	1993	Khuder	Europe (Italy)	Case-control (cancer deaths)	2154 deaths (1674 cancer deaths and 480 referent deaths of all causes) among men aged 35–80 in rural areas surrounding the Viterbo Province between 1980 and 1986	crop	employed > 10 years	10	OR	0.67 (0.29, 1.50)	1.04 (0.43, 2.44)	-
Gallagher [51]	1989	Acquavella	Canada (British Columbia)	PMR	deaths among persons aged 20 and older in British Columbia recorded during 1950 and 1984; 35,668 deaths among male farmers, managers, and laborers compared with all other deaths among the 536,636	mixed	no	111	PMR	0.98 (0.81, 1.18)	-	0.84 (0.68, 1.02)
Gandhi [11]	2014	Agricola	US	Case-control (cancer deaths)	26 brain cancer deaths during 1990–2003 and randomly sampled controls from 43,904 poultry and nonpoultry workers who were members of the United Food and Commercial Worker union between 1949 and 1989 in the US	livestock, crop	no	6	OR	1.6 (0.3, 8.4)	-	-
Gunnarsdottir [52]	1991	Khuder	Europe (Iceland)	Cohort	5922 men registered with the Farmers’ Pension Fund between 1971 and 1980; Icelandic male population was used as a reference	livestock	no	11	SIR	1.28 (0.73, 2.52)	-	0.72 (0.63, 0.82)
Heineman [53]	1995	Khuder	Asia (Shanghai, China)	Cohort	276 cases of brain cancer among women age 30 years or older in the Shanghai Cancer Registry during 1980–1984; the 1982 census and a sample of retirees from the same urban districts were used a reference	crop	grain farmers (higher probability of exposure to pesticides)	3	SIR	2.8 (0.6, 8.0)	6.5 (1.3, 19.1)	-
Howe [54]	1983	Khuder	Canada	Cohort	deaths among 415,201 males in Canada from 1965 to 1969 with data documented on occupation; the Canadian population and the occupational cohort as a whole were used as a reference	not specified	no	4	SMR	1.77 (0.59, 4.22)	-	0.88 (0.79, 0.98)
Inskip [55]	1996	Khuder	Europe (England and Wales)	PMR	all deaths among England and Wales at ages 20–74 during 1979–80 and 1982–90; 62,780 deaths among farmers were compared with those of the general working population	not specified	no	495	PCMR	1.12 (1.03, 1.22)	-	0.96 (0.91, 1.02)
Keller [56]	1994	Khuder	US (Illinois)	Case-control	9514 cancer cases reported to the Illinois state cancer registry between 1986 and 1988 with information on both employment and tobacco use	mixed	no	30	OR	1.39 (0.86, 2.24)	-	-
Kristensen [12]	1996	Khuder	Europe (Norway)	Cohort	cancer occurrence among 246,104 farm holders and spouses in Norway identified through the agricultural census in 1969–1989; the total rural population of Norway was used as reference	livestock, crop	those who purchased pesticides	122	SIR	0.91 (0.77, 1.08)	1.24 (0.89, 1.73)	0.84 (0.71, 1.0)
Lee [57]	2002	Pubmed	US (26 states)	PMR	deaths among from 26 states during 1984–1993; deaths among crop and livestock industry farmers (n = 267,479) were compared to all deaths (~5.7 million)	livestock, crop	no	869	PMR	1.08 (0.93, 1.25)	-	0.90 (0.86, 0.95)
Lee [13]	2005	Pubmed	US (Nebraska)	Case-control	White male cases of histologically confirmed incident cases of glioma aged 21 and older diagnosed between 1988 and 1993 from the Nebraska Cancer Registry or from 11 hospitals (n = 251) and population-based controls either from the general population via random digit dialing or from Medicare files or death certificates (n = 498)	not specified	≥ 55 years farmed	89	OR	1.51 (1.01, 2.27)	3.9 (1.8, 8.6)	-
Mallin [58]	1989	Khuder	US (Illinois)	Case-control (cancer deaths)	deaths from seven cancer sites (n = 10,013) and randomly sampled non-cancer deaths (n = 3198) among black and white men aged 35–74 in Illinois during 1979–1984	not specified	no	70	OR	1.70 (1.23, 2.38)	-	-
Mastrangelo [59]	1996	Khuder	Europe (Padova, Italy)	Cohort	deaths among 2283 male dairy cattle and crop/orchard farmers in Veneto during 1970–1992; the Italian male population was used as a reference	livestock, crop	no	7	SMR	2.36 (1.12, 4.98)	-	-
McLaughlin [60]	1987	Khuder	Europe (Sweden)	Cohort	3394 intercranial gliomas among Swedish males^g,r^ employed in 1960 and followed from 1961 to 1979; the Swedish population was used as a reference	not specified	no	621	SIR	1.10 (1.02, 1.19)	-	-
Menegoz [61]	2002	Pubmed	Multi-national study	Case-control	histologically confirmed meningiomas (n = 330) aged 20–80 years and diagnosed during 1980–1991; 2229 matched (age group, gender, and study center) controls	livestock, crop	no	63	OR	1.05 (0.76, 1.45)	-	-
Milham [62]	1983	Acquavella	US (Washington)	PMR	deaths among 429,926 men during 1950–1979 and 25,066 women excluding homemakers during 1974–1979 in Washington	livestock, crop	yes	69	PMR	1.11 (0.87, 1.40)	-	-
Mills [63]	2006	Pubmed	US (California)	PMR	deaths (n = 3977) among current and former farmworkers who were members of the UFW in California during 1973–2000; the US and Hispanic California populations were used as a reference	not specified	no	18	PMR	0.57 (0.34, 0.90)	-	0.79 (0.73,0.84)
Miranda-Filho [14]	2011	Pubmed	Brazil (Rio de Janeiro)	Case-control (cancer deaths)	brain cancer deaths (n = 2040) among adults aged 18 years and older who resided in Rio de Janeiro during 1996–2005, controls (n = 4140) were selected from among all other causes	mixed	per capita of pesticide sales as an indicator of potential exposure to pesticides	95	OR	1.82 (1.21, 2.71)	1.15 (0.90, 1.47)	-
Musicco [64]	1982	Khuder	Europe (Milan, Italy)	Case-control	patients with glioma (n = 47) and controls (n = 201) with non-neoplastic diseases or benign tumors hospitalized at a neurological institute in Milan, Italy during January 1979 to March 1980; analyzed as matched pairs	not specified	farmers who worked after 1960 when the use of pesticides and fertilizers was higher	17	OR	1.9 (1.23, 4.66)	5.7 (1.66, 25.78)	-
Musicco [15]	1988	Khuder	Europe (Milan, Italy)	Case-control	patients with glioma brain tumors (n = 240) and hospital controls with non-neoplastic neurological diseases without severe psychiatric disorders (n = 742) aged 20–74 years old hospitalized at a neurological institute in Milan between January 1983 and December 1984	not specified	chemical users	61	OR	1.60 (1.06, 2.42)	1.6 (1.04, 2.53)	-
Preston-Martin [37]	1993	Khuder	New Zealand	Case-control	adult male brain cancer patients (n = 1619) and non-brain cancer controls (n = 12,010) obtained from the New Zealand cancer registry between 1972 and 1988	livestock, mixed	no	48	OR	3.23 (2.29, 4.57)	-	-
Rafnsson [65]	1989	Acquavella	Europe (Iceland)	Cohort	5923 farmers registered in the Farmers’ Pension Fund in Iceland between 1977 and 1984; the Icelandic male population was used as a reference	livestock	those born between 1934 and 1945, presumably more pesticide use	7	SMR	1.23 (0.49, 2.53)	1.49 (0.04, 8.32)	0.72 (0.58, 0.89)
Rodvall [66]	1996	Khuder	Europe (Sweden)	Case-control	192 newly diagnosed histologically confirmed cases of glioma and 343 matched controls between the ages 25–74 in central Sweden during 1987–1990	mixed	those with self-estimated exposure to pesticides or weedkillers	33	OR	1.12 (0.65, 1.93)	1.92 (0.81, 4.56)	-
Ronco [39]	1992	Khuder	Europe (Denmark)	Cohort	1970 farmers 15–74 years old registered in the Danish Occupational Cancer Register in 1970; Danish employed population was used as a reference	not specified	no	291	SIR	1.00 (0.90, 1.13)	-	-
Ronco [39]	1992	Khuder	Europe (Italy)	Cohort	cancer deaths in Italian farmers aged 18–74 years between 1981 and 1982; deaths from other causes used as a reference	not specified	no	23	OR	0.54 (0.37, 0.78)	-	-
Ruder [16]	2009	Pubmed	US (Iowa, Michigan, Minnesota, Wisconsin)	Case-control	798 cases ages 18 to 80 years of histologically confirmed glioma diagnosed between 1989 and 1992 residing in Iowa, Michigan, Minnesota, and Wisconsin and identified through medical offices, and 1175 controls without glioma were identified through States’ driver’s license and Medicare data	livestock, crop	pesticides used on farm as child and adult	481	OR	0.85 (0.71, 1.02)	0.61 (0.41, 0.91)	-
Saftlas [17]	1987	Khuder	US (Wisconsin)	PMR	deaths during 1968–1976 among 35,972 white male farmers aged ≥18 years who resided in 69 of 70 Wisconsin counties; US and Wisconsin populations used as a reference	livestock, crop	youngest cohort with a high level of agricultural production herbicide and insecticide exposure	119	PMR	1.10 (0.92, 1.31)	1.15 (0.90, 1.47)	0.92 (0.90, 0.94)
Salerno [18]	2016	Pubmed	Europe (Vercelli, Italy)	Case-control	first hospital admissions for cancer among adults aged 25 to 79 in Vercelli during 2002–2009 (n = 887); controls did not have cancer or had a different type of cancer from the case (n = 11,491)	crop	no	6	OR	4.03 (1.22, 11.89)	-	1.46 (1.23, 1.73)
Schlehofer [67]	2004	Pubmed	International	Case-control	histologically confirmed incident cases (n = 1169) of glioma aged between 20 and 80 years diagnosed during 1980–1991 recruited from either neurosurgical clinics or cancer registries and population-based controls (n = 1981)	not specified	exposure to pesticides	115	OR	0.82 (0.48, 1.40)	1.03 (0.67, 1.57)	-
Schwartz [68]	1986	Acquavella	US (New Hampshire)	PMR	deaths among white males aged 20 years or older in specific occupations between 1975 and 1985 (n = 37,500); the national population was used as a reference	livestock, crop	no	4	PMR	1.33 (0.50, 3.53)	-	0.94 (0.87, 1.01)
Stark [69]	1987	Khuder	US (New York)	Cohort	deaths among 20,833 male farm owners and operators aged 18 and older registered in the Farm Bureau in New York from 1973 to 1984; the census population was used as a reference	mixed	no	12	SMR	1.03 (0.56, 1.75)	-	0.61 (0.55, 0.68)
Stubbs [70]	1984	Khuder	US (California)	PMR	deaths among farm workers and farm owners/managers (n = 14,908) compared with deaths in California during 1978–1979	not specified	no	26	PCMR	1.67 (1.11, 2.41)	-	-
Une [71]	1987	Khuder	US (South Carolina)	PMR	deaths (n = 25,949) among male farmers and non-farmers aged 35 to 84 in South Carolina between 1983 and 1984	not specified	no	9	PMR	0.79 (0.37, 1.70)	-	0.82 (0.74, 0.90)
Walrath [72]	1985	Acquavella	US	Cohort	deaths among 293,958 white males aged 31 to 81 with an active US government life insurance policy in January 1954 in specific occupations compared with all other occupations	not specified	no	27	SMR	1.04 (0.70,1.49)	-	0.90 (0.83, 0.97)
Wigle [73]	1990	Khuder	Canada (Saskatchewan)	Cohort	deaths among 69,513 male farmers in Saskatchewan aged ≥35 years during 1971–1985^m^	mixed	no	96	SMR	1.03 (0.84, 1.25)	-	0.81 (0.78, 0.84)
Wiklund [38]	1986	Acquavella	Europe (Sweden)	Cohort	cancer incidence among 604,103 men and women working in agriculture or forestry in Sweden during 1961–1979	mixed	no	819	RR	1.04 (0.96,1.12)	-	0.82 (0.81, 0.83)
Wiklund [19]	1994	Khuder	Europe (Sweden)	Cohort	50,682 women who worked at least 20 h per week in agriculture as reported in the 1970 Swedish Census, followed from 1971 to 1987	mixed	birth cohort 1935+ with the rationale that younger persons tended to use more pesticides	189	SIR	1.05 (0.91, 1.22)	0.85 (0.43, 1.5)	0.85 (0.82, 0.87)
Wiklund [74]	1995	Khuder	Europe (Sweden)	Cohort	140,208 men who worked at least 20 h per week in agriculture as reported in the 1970 Swedish Census, followed from 1971 to 1987	mixed	birth cohort 1935+ with the rationale that younger persons tended to use more pesticides	488	SIR	1.00 (0.91, 1.09)	1.07 (0.77, 1.46)	0.80 (0.78, 0.81)
Wingren [75]	1992	Khuder	Europe (Sweden)	Case-control (cancer deaths)	570,979 deaths among men age 45 and older in Sweden during 1950–1982; brain cancer deaths compared with deaths from nonmalignant cancers and noncardiovascular disorders	not specified	no	11	OR	4.80 (2.70, 8.50)	-	-
Zheng [76]	2001	Pubmed	US (Iowa)	Case-control	412 histologically confirmed cases of glioma aged 40–85 years identified by the Iowa state health registry during 1984–1987, 2434 population-based controls without a previous cancer diagnosis identified through drivers licenses and Medicare records	not specified	≥10 years duration of employment	111	OR	1.33 (1.05, 1.70)	1.27 (0.92, 1.76)	-

**Table 2 cancers-13-04477-t002:** Pooled risk estimates from meta-analyses of farming and brain cancer by selected characteristics of studies.

Characteristics	Subcategory	Number of Studies	Pooled Risk Estimate (95% CI)	*p*-Value Homogeneity
Overall	-	52	1.13 (1.06, 1.21)	<0.0001
Year of Publication				
	Prior to 1990	19	1.12 (1.04, 1.21)	0.15
	Between 1990 and 1999	23	1.12 (1.01, 1.25)	<0.0001
	2000 or later	10	1.13 (0.92, 1.39)	<0.0001
Study Design				
	Case-control	20	1.39 (1.10, 1.75)	<0.0001
	Cohort	20	1.01 (0.95, 1.08)	0.005
	Proportional Mortality Ratio	12	1.10 (1.02, 1.17)	0.135
Region				
	United States	22	1.13 (1.03, 1.23)	0.02
	Europe/United Kingdom	21	1.08 (0.98, 1.17)	<0.0001
	Canada	4	0.92 (0.66, 1.26)	0.013
Sex				
	Male	41	1.14 (1.07, 1.22)	<0.0001
	Female	13	1.11 (0.90, 1.36)	<0.0001
Race				
	White	16	1.21 (1.05, 1.39)	<0.0001
	Non-white	5	1.16 (0.92, 1.46)	0.25
Type of Farmer				
	Crop	16	1.13 (0.97, 1.30)	<0.0001
	Livestock	13	1.34 (1.18, 1.53)	<0.0001
Farming Duration				
	Less than 10 years	3	1.42 (0.63, 3.21)	0.11
	10 years or more	4	1.72 (1.15, 2.58)	0.03
Type of Brain Cancer				
	Glioma	8	1.11 (0.95, 1.31)	0.05
Source of Exposure Classification				
	Personal Interview, Self-Report	9	1.15 (0.88, 1.51)	0.006
	Registry, Administrative List	26	1.12 (1.01, 1.24)	<0.0001
	Death Certificate	16	1.13 (1.02, 1.24)	<0.0001
Quality Tier				
	Good	12	1.40 (1.09, 1.81)	<0.0001
	Fair	20	1.04 (0.94, 1.15)	0.001
	Poor	20	1.07 (0.99, 1.17)	<0.0001
More highly exposed subcohort		19	1.15 (1.0, 1.32)	0.01
Documented exposure to pesticides		10	1.22 (1.10, 1.35)	0.04

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
