# Peer review of "Farming, Pesticides, and Brain Cancer: A 20-Year Updated Systematic Literature Review and Meta-Analysis"

_cancers, 2021, doi:10.3390/cancers13174477_

Round 1
Reviewer 1 Report
I read the proposed manuscript with great interest. Authors demonstrated that the synthesis of epidemiologic studies supports an increased risk of brain cancer from farming with its potential for exposure to chemical pesticides.
The manuscript is written in clear language, and references are provided in full. I believe that it can be published in Cancer journal with minor revisions:
Line 92-94 – Why they didn’t use new and old information together?
Line 100 – better: European Union and Great Britain, China, Brazil, New Zealand, and International.
Line 319-320 – pesticides (insecticides, herbicides, fungicides, disinfectants, and means for the destruction of rats and mice
Second paragraph page 15, Lines 340-341 - expand the paragraph, mention to the list such chemicals widely generated in agriculture as formaldehyde, ammonia, hydrogen sulfide, phenols, and others.
It should also be noted here that pesticides used in agriculture in different countries have different toxicity and in the future, the authors will try to trace the influence of different chemical classes (organochlorine, organophosphorus, neonicotinoids, carbamates, phenyl pyrazoles, etc.)
Also, in different countries, completely different pesticides are used for the same purposes (from the newest to those discontinued in production, but still used in agriculture in some countries.)
But I repeat, this is a very good manuscript and I would recommend it for publication.
Author Response
Line 92-94 – Why they didn’t use new and old information together?
- Response: Thank you to the reviewer for this question. If we are understanding the inquiry, generally a “newer” study would have included the “older” study data, so we did include both “new” and “old” information together by virtue of selecting the more recently published article.
Line 100 – better: European Union and Great Britain, China, Brazil, New Zealand, and International.
- Response: Thank you to the reviewer for this suggestion of how to improve the clarity of the methods. Many of the articles included were published prior to the formation of the EU so this descriptor would not be accurate. We do edit line 100 as suggested to specify China (instead of Asia), Brazil, New Zealand and International.
Line 319-320 – pesticides (insecticides, herbicides, fungicides, disinfectants, and means for the destruction of rats and mice
- Response: We thank the reviewer for this suggestion. We have added “to destroy rodents” to line 319.
Second paragraph page 15, Lines 340-341 - expand the paragraph, mention to the list such chemicals widely generated in agriculture as formaldehyde, ammonia, hydrogen sulfide, phenols, and others.
- Response: We have added such a statement to this paragraph.
It should also be noted here that pesticides used in agriculture in different countries have different toxicity and in the future, the authors will try to trace the influence of different chemical classes (organochlorine, organophosphorus, neonicotinoids, carbamates, phenyl pyrazoles, etc.)
Also, in different countries, completely different pesticides are used for the same purposes (from the newest to those discontinued in production, but still used in agriculture in some countries.)
- Response: Excellent comment. We thank the reviewer for mentioning this. It was our intention to discuss these additional points but they were inadvertently not included. We have added this information to lines 337-338.

Reviewer 2 Report
- Abstract part should be improved as few lines are not clear.
- write Peoples after white and black in introduction part
- Effect of chemical pesticides on brain tumor should be elaborated with latest refrences
- Discussion part should be improved by including the the role of sustainable agriculture in controlling the cancer
Author Response
1.write Peoples after white and black in introduction part
- Response: So added.
2.Effect of chemical pesticides on brain tumor should be elaborated with latest references.
- Response: We thank the reviewer for this helpful suggestion. We have included several of the more recent studies in the references (line 54).
3.Discussion part should be improved by including the the role of sustainable agriculture in controlling the cancer
- Response: We thank the reviewer for this comment. We have added such a statement to our Conclusion.
